# Molecular Insights into Diapause Mechanisms in *Telenomus remus* for Improved Biological Control

**DOI:** 10.3390/insects16040393

**Published:** 2025-04-08

**Authors:** Guojie Yu, Longyu Sheng, Zhongyue Zhang, Qi Zou, Xinxin Lai, Yan Tang, Yuyao Li, Jia Liu, Hao Yan, Xianglin Xie, Fei Hu, Zengxia Wang

**Affiliations:** 1College of Resource and Environment, Anhui Science and Technology University, Fengyang 233100, China; yuguojie2001@163.com (G.Y.); 17855025127@163.com (L.S.); zhangzhongyue0921@163.com (Z.Z.); zouqi1028@163.com (Q.Z.); 19556954801@163.com (X.L.); ty20000314@163.com (Y.T.); 15555086186@163.com (Y.L.); 19556957820@163.com (J.L.); yan475800@163.com (H.Y.); xxl342201@163.com (X.X.); 2Institute of Plant Protection and Agro-Products Safety, Anhui Academy of Agricultural Sciences, Hefei 230031, China; hufly0224@163.com; 3College of Agriculture, Anhui Science and Technology University, Fengyang 233100, China; 4Anhui Engineering Research Center for Smart Crop Planting and Processin Technology, Anhui Science and Technology University, Fengyang 233100, China

**Keywords:** diapause, *Telenomus remus*, *Spodoptera frugiperda*, transcriptome analysis, photoperiod, calcium signaling, pest control, sustainable agriculture

## Abstract

This study focuses on understanding and improving the use of a tiny beneficial insect, *Telenomus remus* (Nixon), on controlling an invasive pest called *Spodoptera frugiperda* (Smith), which destroys crops like corn. These insects, known as parasitoid wasps, lay their eggs inside the pest’s eggs, preventing them from hatching and causing crop damage. However, one major challenge is that these wasps have a short lifespan and are difficult to store for long periods, making it harder to use them effectively in pest control programs. To address this, researchers studied a remarkable state called diapause, where the insects temporarily stop developing and can survive longer under controlled conditions. By exposing the wasps to specific light and temperature settings, they entered this dormant state. Using advanced genetic techniques, the study identified key genes and processes that control diapause. Two important genes were found to play major roles in maintaining this state. These findings help scientists better understand how to manage and store these wasps, allowing for more efficient pest control. This study explores the diapause mechanisms of *T. remus* and its application in biological control, where reproductive diapause is used to control the timing of natural enemy release, delaying their release until the pests begin laying eggs, thereby improving biological control effectiveness.

## 1. Introduction

*Spodoptera frugiperda* (Smith), belonging to the order Lepidoptera and the family Noctuidae, is a major migratory agricultural pest of global significance [1]. Biological control, mainly using natural enemy insects, is an effective strategy for the sustainable management of *S. frugiperda*, with promising applications. *Telenomus remus* (Nixon), a member of the order Hymenoptera and the family Platygastridae, has been found parasitizing *S. frugiperda* in Guangdong, Hainan, Guizhou, Anhui, and other provinces of China following the pest’s invasion. It has been proven to be a dominant egg parasitoid of *S. frugiperda* [2,3,4,5,6].

The large-scale artificial propagation of *T. remus* is a prerequisite for its practical application in controlling *S. frugiperda*. However, *T. remus*’s short lifespan and limited viability under low-temperature storage often results in its death before the pest outbreak or the absence of available natural enemies when the pest occurs. Foerster et al. found that the fecundity of *Telenomus podisi* (Ashmead) decreased by approximately 80% after cold storage [7]. Bayram et al. measured the laboratory performance of emerged parasitoids and compared it with that of the unstored (control) parasitoids. The results showed that cold storage had a significant adverse effect on the mean adult emergence rate. Additionally, the mean parasitism rate of host eggs by emerged parasitoids decreased, and the sex ratio of F1 progeny became more male-biased as the storage duration increased [8]. In this study, reproductive diapause is described as a process that reduces metabolic rate, thereby slowing down individual aging, extending adult lifespan, and enabling the parasitoid to retain its reproductive capacity when the pest outbreak occurs. Diapause plays a significant role in the propagation and application of tiny parasitoids.

In recent years, with the deepening of insect diapause research, studies on the diapause of parasitoids have also progressed. Currently, research on parasitoid research mainly focuses on species from Braconidae, Ichneumonidae, and Trichogrammatidae [9]. Most parasitoids enter diapause induced by low temperature or photoperiod, primarily as pupae or larvae. For instance, *Aphidius gifuensis* (Ashmead) enters diapause as pupae and final instar larvae induced by low temperature and short photoperiod [10,11]. In contrast, *Nasonia vitripennis* (Walker) enters diapause as final instar larvae induced by low temperature and short photoperiod [12,13]. Reproductive diapause is a dormant state in insects during their life cycle, typically occurring under adverse environmental conditions. This dormant state pauses the insect’s reproductive activities, allowing them to resume reproduction at a more favorable time, usually when the climate or food resources are more abundant. Reproductive diapause differs from other types of diapause (such as developmental diapause or behavioral diapause) in that it involves slowing down or ceasing the insect’s reproductive system activity. Research on the genus Telenomus is limited, with only Legault et al. [14] discovering that Telenomus has reproductive diapause characteristics. Building on this, our experiments have shown that the reproductive diapause of *T. remus* includes both the egg and adult stages, and it is a facultative diapause. Given this, researchers in our laboratory have explored the diapause induction conditions for *T. remus* and found that it exhibits reproductive diapause under 0L:24D photoperiod conditions. This discovery provides the possibility for the industrial production and application of *T. remus*. Storing *T. remus* in a diapause state and then breaking the diapause before field release does not affect its viability and ensures high survival rates. This approach can regulate the growth and development of the parasitoid, extend product shelf life, prolong control efficacy, and enhance resistance and reproductive capacity [15].

Diapause induction in natural enemy insects mainly relies on temperature and photoperiod conditions, which is time-consuming and inefficient, hindering the large-scale propagation of natural enemies. In recent years, with the advancement of next-generation sequencing technology, significant progress has been made in studying insect diapause through transcriptome sequencing. In studies on the diapause mechanism of *A. gifuensis*, high-throughput transcriptome sequencing was used to identify diapause-associated genes, obtaining 458 diapause-associated genes [16]. Analysis of diapause-related genes in *Delphastus catalinae* (Horn) revealed 571 differentially expressed genes expressed explicitly during diapause, involving 116 KEGG enriched pathways [17]. Research has shown that calcium-binding protein (CBP) and arrestin (ARR) are essential in light signal transduction. CBP is a protein involved in calcium ion regulation during various physiological processes and is closely related to cellular signal transduction and environmental adaptation. ARR, on the other hand, is a typical light receptor-related protein known to play a significant role in light perception and response in both plants and insects. This experiment conducted transcriptome sequencing the normal developmental and diapause states of the parasitoid *T. remus*. KEGG pathway enrichment analysis was performed to identify genes associated with diapause. Since the diapause samples were induced by photoperiod, two key genes in the light signal transduction pathway (CBP and ARR) were screened and identified. Bioinformatics analysis and differential expression validation of these key genes were conducted to explore the molecular mechanisms of diapause. The goal was to identify genes that play a key role in diapause regulation in *T. remus*, providing new insights for inducing diapause in this species and improving its application in the control of pests like the *S. frugiperda* and other lepidopteran pests. In this experiment, transcriptome sequencing of *T. remus* in normal and diapause states was conducted to perform KEGG pathway enrichment analysis of diapause-associated genes, screen diapause-associated genes, and conduct bioinformatics analysis and differential expression validation of key genes. The aim is to elucidate the molecular mechanism of diapause in *T. remus*, identify key genes involved in diapause regulation, and provide a new approach for diapause induction, thereby enhancing the application of *T. remus* in controlling *S. frugiperda* and other Lepidoptera pests.

## 2. Materials and Methods

### 2.1. Test Insects

The *T. remus* and its host *S. frugiperda* used in this experiment were collected from a cornfield near Fengyang, Chuzhou, Anhui Province (117.56° E, 32.86° N) in October 2020. They were reared for multiple generations indoors to establish an experimental population. *S. frugiperda* larvae were fed with fresh corn leaves and an artificial diet in a rearing room with a temperature of 28 (±1) °C and relative humidity of 80% (±5%). A photoperiod of L:D = 14:10. After pupation, the pupae were placed in rearing cages (diameter 27 cm, height 40 cm) for eclosion. The emerged adults were fed with 10% honey water, and fresh egg masses were collected daily for the experiment. *T. remus* was reared in rearing tubes (diameter 2 cm, length 8 cm) with 1-day-old *S. frugiperda* eggs as the host. The adults were fed with 10% honey water daily, under rearing conditions of 24 (±1) °C, 70% (±10%) relative humidity, and a photoperiod of L:D = 14:10. Mated females that had not laid eggs for 24 h were selected for the experiment.

### 2.2. Diapause Induction

According to Legault’s method for determining reproductive diapause in parasitic wasps of the genus Telenomus [14], female wasps that fail to parasitize host eggs are classified as being in diapause. Based on previous research results, *T. remus*-parasitized *S. frugiperda* egg masses were incubated under a temperature of 24 (±1) °C), relative humidity of 80% (±5%), and photoperiod 0L:24D. *T. remus* remained in complete darkness from the egg stage to adult emergence. After emergence, each female was given 50 fresh one-day-old *S. frugiperda* eggs for 24 h to parasitize. If a female failed to parasitize the eggs, another batch of 50 one-day-old eggs was provided for an additional 24-h parasitization attempt. Females that failed to parasitize in both attempts were classified as being in diapause, with a diapause induction rate of 70%. The rearing conditions for the control group were temperature 24 (±1) °C, relative humidity 70% (±10%), and photoperiod L:D = 14:10. Both the experimental group and the control group used 50 one-day-old egg masses for the experiment, with three biological replicates for each group.

To terminate diapause, the diapausing females were exposed to a photoperiod of 14L:10D. If the females regained their ability to parasitize *S. frugiperda* eggs after exposure, they were classified as having exited diapause. Under 14L:10D conditions, 48 h of light exposure achieved a diapause termination rate of 78%.

In the experiment, 50 diapause and 50 non-diapause females were selected for subsequent transcriptome sequencing analyses, with three biological replicates for all samples. Collected samples were flash-frozen in liquid nitrogen and stored at −80 °C for further use.

### 2.3. Total RNA Extraction and Detection

Total RNA was extracted from diapause and non-diapause samples of *T. remus* using the Trizol method [18]. The integrity of the RNA was checked by 1% agarose gel electrophoresis, and RNA purity and quality were assessed with a Nanodrop ND-2000. Qualifying RNA samples were sent to Novogene (Beijing, China) for transcriptome sequencing analysis.

### 2.4. Transcriptome Assembly and Annotation

Raw data from the Illumina HiSeq sequencing platform were filtered to remove adaptor sequences, low-quality reads, and reads with undetermined base information, yielding high-quality clean reads. These were assembled de novo using Trinity (Trinity-v 2.5.1) to obtain unigene sequences. The quality of the assembly was evaluated with BUSCO software (BUSCO v5.4.1) using Trinity fasta files, assessing accuracy and completeness based on GC content and unigene integrity. The resulting unigenes were annotated by BLAST 2.2.26 against seven databases: Nr, Nt, Pfam, KOG/COG, GO, KEGG, and Swiss-Prot.

### 2.5. Screening of Diapause-Associated Genes and KEGG Pathway Analysis

Using the unigenes as a reference, gene expression levels for each sample were calculated with FPKM (Fragments Per Kilobase of transcript per Million mapped reads). Differential genes between the non-diapause and diapause groups were identified with a q-value ≤ 0.005 and |log2FoldChange| ≥ 1.0. Genes upregulated or downregulated in the diapause group compared to the control group were defined as diapause-associated genes. These gene sequences were extracted for further analysis. The sequences were formatted in FASTA and analyzed using the KAAS (KEGG Automatic Annotation Server, http://www.genome.jp/kegg, accessed on 4 April 2023) online tool for KEGG pathway enrichment.

### 2.6. Identification and Phylogenetic Analysis of Light Signal Transduction Genes

As the diapause samples were induced by photoperiod, two key genes in the light signal transduction pathway (CBP and ARR) were screened and identified. Protein sequence characteristics and domains were analyzed. Amino acid sequences of candidate genes were aligned using MAFFT (https://www.ebi.ac.uk/Tools/msa/mafft/, accessed on 4 April 2023) with the FFT-NS-I iterative method and JT200 scoring matrix. Phylogenetic trees were constructed using the neighbor-joining method in MEGA 5.0 and edited with Figtree.

### 2.7. Quantitative Real-Time PCR Validation of Light Signal Transduction Gene Expression

The expression of CBP and ARR genes in diapause and non-diapause samples of *T. remus* was analyzed using a fluorescence qPCR (quantitative Polymerase Chain Reaction) kit following the manufacturer’s instructions. The qPCR reagent kit we used is [TB Green^®^ Premix Ex Taq™II (Tli RNaseH Plus)], provided by [TAKARA, Kusatsu, Japan]. Each sample had three biological replicates. The 20 μL reaction mixture contained 1 μL of cDNA template, 0.4 μL each of 10 mmol/L forward and reverse primers, 10 μL of 2×SYBR^®^ Green Mix dye, and 7.8 μL of RNase-free water. The reaction program was 95 °C for 30 s; 95 °C for 5 s, 60 °C for 30 s, 40 cycles; 95 °C for 15 s, 60 °C for 1 min. The RPL12 and RPS2 genes of *T. remus* were used as reference genes. Specific primers for CBP, ARR, and reference genes were designed using Primer Premier 5.0 (Table 1). The relative expression levels of genes were calculated using the 2^−△△Ct^ method, and significant differences were analyzed with SPSS 25.0. Graphical analyses were performed using GraphPad Prism 9.5.0 software.

## 3. Results

### 3.1. Transcriptome Sequencing and Data Assembly of T. remus

Transcriptome sequencing was performed on *T. remus* during both the non-diapause and diapause periods. After removing low-quality reads and those containing adapters, a total of 44,241,627 clean reads were obtained. The data showed that the GC content ranged from 43.76% to 44.49%, with Q20 values consistently above 97.19% and Q30 values consistently above 92.26%. Using Trinity software to assemble and cluster the clean reads, a total of 16,232 unigenes were obtained, with an average length of 1921 bp and an N50 length of 3540 bp. The data indicates that the transcriptome assembly of *T. remus* was relatively successful (Table 2).

### 3.2. Functional Annotation of Unigenes in T. remus

As shown in Figure 1, the obtained unigene sequences were compared against seven major databases using BLAST sequence alignment, resulting in a total of 16,232 annotations. Specifically, 11,730 annotations were obtained from the NR database, 8182 from the NT database, 7992 from the KO database, 6611 from the KOG database, 9389 from the PFAM database, 9388 from the GO database, and 9606 from the Swiss-Prot database.

### 3.3. Functional Classification of Unigenes in T. remus

Through GO database analysis, a total of 9388 homologous proteins were annotated, accounting for 57.84% of the total unigenes in the transcriptome. These annotated unigenes were categorized into three major categories and 41 functional subgroups: molecular function, cellular component, and biological processes. Among these, the largest number of unigenes were involved in molecular functions, as shown in Figure 2.

### 3.4. Functional Annotation of T. remus in KEGG Database

According to the KEGG database comparison results, a total of 7992 genes were annotated, accounting for 49.23% of the total unigenes. These genes are involved in 299 metabolic pathways. The pathway with the most unigenes involved is the ribosome metabolic pathway, with 273 unigenes, accounting for 3.4%. This is followed by the protein processing in the endoplasmic reticulum pathway with 178 unigenes (2.23%), and the oxidative phosphorylation pathway with 177 unigenes (2.21%). Table 3 lists the top 20 enriched KEGG metabolic pathways. For a detailed analysis of KEGG pathways related to diapause-associated genes in *T. remus*, please refer to the attachment.

### 3.5. Screening of Differentially Expressed Genes

The diapause group and non-diapause group are denoted by the letters D and CK, respectively. Using the criteria of qval ≤ 0.005 and |log2FoldChange| ≥ 1.0, a total of 16,232 unigenes were screened to identify significantly differentially expressed genes between the diapause and non-diapause groups in *T. remus*. The results showed that, compared to the control group, 212 genes were upregulated and 2430 genes were downregulated in the diapause group, as shown in Figure 3.

Through the pathway analysis of differentially expressed genes during diapause and non-diapause periods, 617 differentially expressed genes were found to be involved in 284 pathways. Among these, six pathways showed significant enrichment during diapause (padj < 0.05). The most enriched pathway was the ribosome metabolic pathway, with 108 differentially expressed genes enriched in this pathway, accounting for 17.50%. The next most enriched pathway was oxidative phosphorylation, accounting for 14.74%. These results suggest that *T. remus* undergoes significant changes in ribosome and oxidative phosphorylation pathways during diapause.

### 3.6. Identification and Phylogenetic Analysis of Light Signal Transduction Genes in T. remus

In this experiment, diapause in *T. remus* was induced by photoperiod. Analysis of transcriptome sequencing data revealed several differentially expressed genes enriched in light signal transduction pathways such as the calcium signaling pathway (ko04020) and phototransduction pathways (ko04744, ko04745).

Screening of light signal transduction genes in these pathways identified four CBP genes and six ARR genes. To understand the changes and roles of these two types of genes during the induction of diapause in *T. remus*, this study analyzed the sequences of CBP and ARR genes and performed phylogenetic analysis. The aim is to clarify the diapause regulation mechanism in *T. remus*.

#### 3.6.1. CBP Sequence Analysis

Analysis of the CBP gene sequences from four *T. remus* revealed that the ORF length of the *Trem*CBP gene ranges from 588 to 1761 bp, encoding 196 to 587 amino acids (Table 4). Functional domain analysis of the encoded protein (Figure 4) showed that members of the CBP family contain one or more typical EF-hand domains, which are the primary structures for calcium ion binding. Phylogenetic tree analysis (Figure 5) demonstrated that CBP family genes are mainly divided into four major clusters. CBP1 clusters with the sarcoplasmic calcium-binding protein (SCP) from *Apis mellifera* and *Leptopilina heterotoma*. CBP2 shows the closest relationship to the SPARC-related modular calcium-binding protein (SMOC) from *Osmia lignaria*. CBP3 clusters with the 45 kDa calcium-binding protein (Cab45) from *A. mellifera*, and CBP4 clusters with the calmodulin (CAM) family, showing the closest relationship to *Monomorium pharaonis* and *A. mellifera*. Previous studies have found that these four types of proteins play roles in the regulation of calcium ion-mediated light signal transduction.

As shown in Figure 6, expression analysis of CBP genes in diapausing and non-diapausing *T. remus* revealed that all four types of CBP genes are expressed in both conditions. However, CBP1 and CBP2 were significantly downregulated after diapause treatment (F (3, 16) = 74.83, *p* < 0.01). These results suggest that CBP1and CBP2 genes may play distinct roles during the diapause process of *T. remus*.

#### 3.6.2. ARR Sequence Analysis

Analysis of the six ARR genes in *T. remus* revealed that their ORFs range from 336 to 1521 bp, encoding 112 to 507 amino acids. Protein domain predictions (Figure 7) indicated that most ARR genes in *T. remus* contain both N-terminal and C-terminal Arrestin domains.

Phylogenetic tree analysis (Figure 8) showed that the ARR gene family in *T. remus* clusters into four main branches. Among them, ARR1 groups with the *Beta*-arrestin-1 family from various insects, ARR2 has the closest relationship to ARRDC2 from Camponotus floridanus, ARR3 shows the closest relationship to Cephus cinctus, and ARR4, ARR5, and ARR6 form a cluster with the ARRDC17 family.

As shown in Figure 9, expression analysis of the ARR gene family in diapausing and non-diapausing *T. remus* revealed significant differences in expression levels among the three ARR genes under different treatments. ARR2 was expressed at low levels, indicating low-abundance expression. ARR1 and ARR5 were all significantly downregulated after diapause treatment (F (5, 24) = 68.94, *p* < 0.01), while ARR*4* was significantly upregulated after diapause treatment, and ARR*6* showed no significant changes in expression.

## 4. Discussion

Diapause is an adaptive strategy for insects to cope with environmental conditions, generally occurring at various stages of insect development [19]. Based on the developmental stage, diapause can be categorized into egg diapause, larval diapause, pupal diapause, and adult diapause [20]. In this study, transcriptome sequencing was conducted using whole-body *T. remus*, with samples for the diapause group selected by inducing diapause from the egg stage to unmated adult females under conditions of 24 °C and a photoperiod of 0L:24D. Sampling of non-diapause and diapause female adults of *T. remus* mainly relied on photoperiod regulation. Sampling of non-diapause and diapausing female adults of *T. remus* primarily relied on photoperiod regulation.

Transcriptome technology has advanced the study of insect diapause-related genes, providing theoretical support for uncovering the mechanisms of insect life activities [21,22,23]. Bao et al. [24] conducted a time-series transcriptome analysis to reveal the molecular mechanisms underlying diapause termination in *Chilo suppressalis* (Walker), identifying 1056 shared DEGs (differentially expressed genes) when comparing WSG2, WSG3, and WSG4 with WSG1. These DEGs reflected differences in gene expression between the diapause and diapause termination stages of *C. suppressalis*. In this study, a comparison between the diapause and control groups revealed 212 upregulated genes and 2430 downregulated genes. KEGG pathway analysis identified 617 differentially expressed genes involved in 284 pathways, with six pathways showing significant enrichment during the diapause period.

During the diapause induction, insects respond to, receive, and process environmental signals to determine whether to enter diapause or continue development [19,25]. *T. remus* primarily enters diapause through photoperiod induction; thus, light signal transduction plays a critical regulatory role during this stage. Calcium ions (Ca^2+^), as secondary messengers in cells, are involved in numerous intracellular signal transduction processes, including light signal transduction. Previous studies have highlighted the key role of Ca^2+^ in plant light signal transduction [26] and its essential function in insect visual signal transduction [27,28]. Calcium ions are also highly likely to participate in diapause induction in insects by regulating light signal transduction. In this study, qRT-PCR results showed that the expression levels of CBP1 and CBP2 were significantly downregulated after diapaused individuals’ diapause treatment, suggesting that these genes might be suppressed during diapause. According to Fu et al. [29], the putative juvenile hormone diol kinase (LdJHDK) in the *Leptinotarsa decemlineata* (Say) exhibits structural similarities to CBP. Through qRT-PCR analysis, they discovered that the expression of the LdJHDK gene was downregulated during diapause. This downregulation is related to regulating juvenile hormone (JH) levels, indicating that this gene plays a key role in hormone metabolism and insect development during diapause. This finding is consistent with the conclusions of our current study. In diapausing individuals of *Helicoverpa armigera* (Hubner), CBP were significantly downregulated [30], consistent with our findings. CAM, a CBP, regulates various signal transduction processes in organisms and is particularly important in the visual system of Drosophila, where it terminates light signal transduction by phosphorylating rhodopsin ARR proteins [31,32]. Studies have also found that CAM regulates the diapause process of *L. decemlineata* [23]. The diapause type of the *L. decemlineata* belongs to hibernation diapause. In this study, differentially expressed genes were enriched in the calcium signaling pathway, suggesting that calcium ions are likely involved in regulating diapause induction in *T. remus* and play a role in light signal transduction during this process.

In addition, sequencing results also revealed the specific expression of the ARR gene in the “Phototransduction” pathway. ARR is known to be involved in light signal transduction in insects [33]. The diapause of Drosophila is primarily characterized as reproductive diapause, mainly occurring in the adult stage (adult diapause). In Drosophila, ARR is a critical molecule in light signal transduction, capable of regulating the activity of rhodopsin and working in conjunction with transient receptor potential (TRP) channels to modulate light responses [34]. This indicates that ARR has a universal role in photoreception and signal transduction. This study hypothesizes that the ARR gene may also participate in light signal transduction during the diapause induction process in *T. remus*.

In investigating light signal transduction-related genes during diapause in *T. remus*, we found that ARR and calcium-binding proteins may play key roles in regulating the transition between light signaling and the diapause state. Wu et al. [35] demonstrated that ARR can integrate multiple signaling pathways through direct interaction with CAM, providing strong evidence for the multifunctionality of ARR in diapause regulation. During the diapause process in *T. remus*, ARR may regulate light signals by interacting with CBP, thereby influencing the initiation and maintenance of diapause.

## 5. Conclusions

The significant downregulation of CBP1 and CBP2 may reflect the regulatory mechanism of the phototransduction pathway during diapause in insects. Specifically, by inhibiting calcium signaling, insects may reduce their responsiveness to external light stimuli while simultaneously decreasing metabolic activity to meet the energy demands of diapause. This regulatory strategy may enhance insect survival under unfavorable environmental conditions and enable a rapid resumption of physiological activity when conditions become favorable. Combined with the differential expression of the ARR gene, this study suggests that the phototransduction and calcium signaling pathways work together in diapause regulation, influencing insect metabolic adjustment and environmental adaptability.

*T. remus*, a dominant natural enemy of noctuid pests such as *S. frugiperda*, has an effective control impact on these pests and plays an important role in the sustainable management of pest populations. This study analyzes the molecular regulatory mechanisms of diapause in *T. remus*, elucidating the scientific mechanisms behind biological population regulation. By relying on artificial control of diapause and adjusting the release of the parasitic wasp based on the occurrence of pests, this research aims to improve the utilization efficiency of natural enemy insect products and contribute positively to enhancing their pest control effectiveness.

## Figures and Tables

**Figure 1 insects-16-00393-f001:**
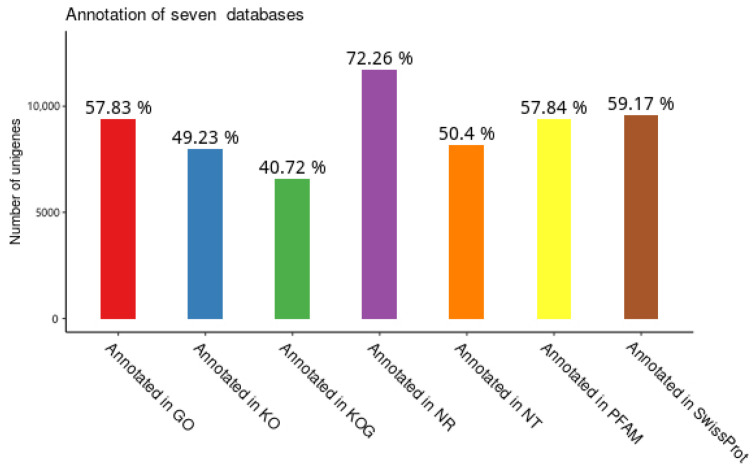
Statistics of Unigene Annotations.

**Figure 2 insects-16-00393-f002:**
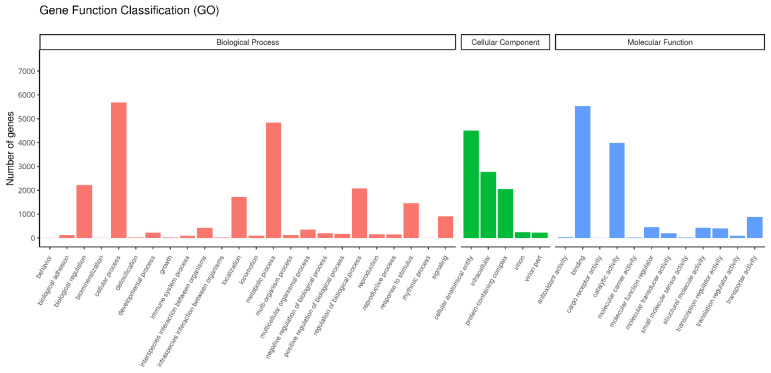
Functional Classification of Unigenes in *T. remus*. Within the molecular function category, the most abundant unigenes were those involved in binding functions (5524) and catalytic activity (3990). In the cellular component category, the unigenes were predominantly involved in cell anatomical entities (4499), followed by intracellular components (2773) and protein complexes (2061). In the biological processes category, the main functions were cell processes (5689) and metabolic processes (4835).

**Figure 3 insects-16-00393-f003:**
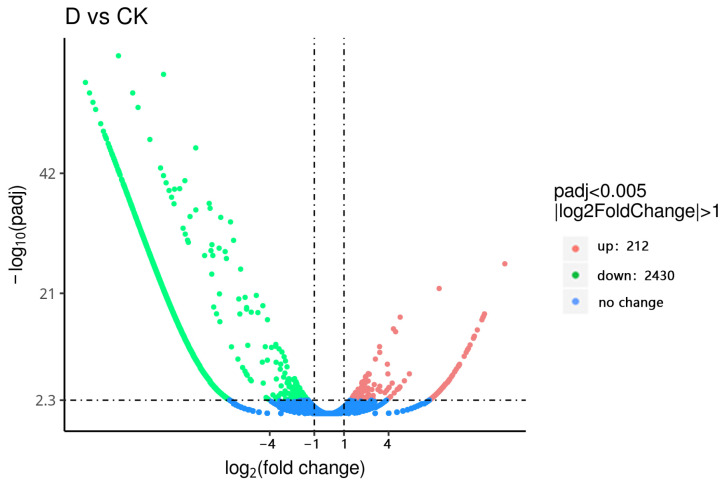
Differentially Expressed Genes in *T. remus*.

**Figure 4 insects-16-00393-f004:**
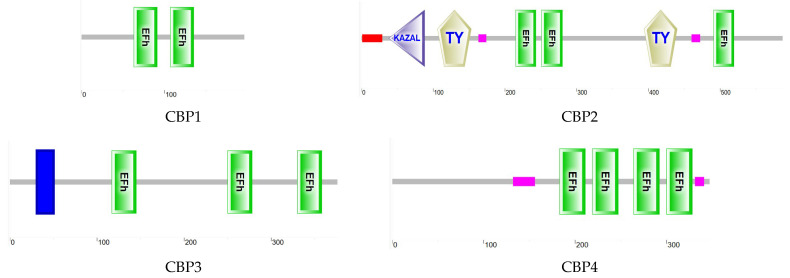
CBP Protein Domains of *T. remus*.

**Figure 5 insects-16-00393-f005:**
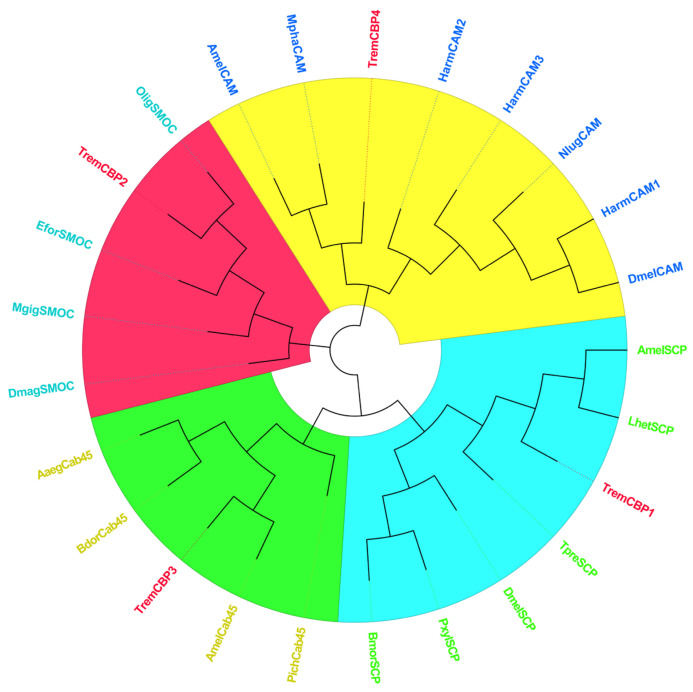
Phylogenetic Tree of CBP Protein Sequences in *T. remus*, Comparative species include: [*Apis mellifera*] Cab45 XP_026298353.1, [*Pseudobagrus ichikawai*] Cab45 GAA6095069.1, [*Bactrocera dorsalis*] Cab45 JAC46915.1, [*Aedes aegypti*] Cab45 XP_001654735.1, [*Helicoverpa armigera*] CAM XP_021192593.1, [*Helicoverpa armigera*] CAM XP_021192591.1, [*Helicoverpa armigera*] CAM XP_063899395.1, [*Nilaparvata lugens*] CAM XP_022191617.1, [*Drosophila melanogaster*] CAM AHN56135.1, [*Apis mellifera caucasica*] CAM KAG6803258.1, [*Monomorium pharaonis*] CAM XP_028045948.1, [*Apis mellifera*] SCP XP_623244.1, [*Trichogramma pretiosum*] SCP XP_014227739.1, [*Plutella xylostella*] SCP XP_011554969.1, [*Bombyx mori*] SCP XP_037874355.1, [*Leptopilina heterotoma*] SCP XP_043481086.1, [*Drosophila melanogaster*] SCP EAA46049.1, [*Osmia lignaria*] SMOC XP_034171452.1, [*Daphnia magna*] SMOC JAK86345.1, [*Euwallacea fornicatus*] SMOC XP_066151365.1, [*Magallana gigas*] SMOC XP_065943295.1.

**Figure 6 insects-16-00393-f006:**
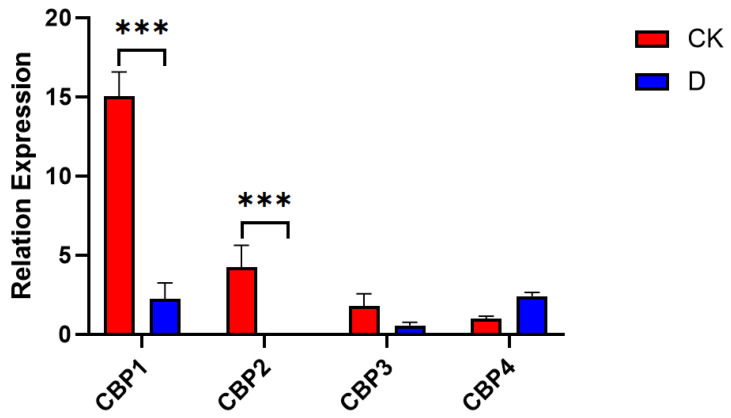
Expression Analysis of CBP Genes in Diapausing and Non-Diapausing *T. remus*. (*** represents a significant difference *p* < 0.01, “CK” refers to the normally reared *T. remus*, and “D” refers to *T. remus* that underwent diapause treatment.).

**Figure 7 insects-16-00393-f007:**
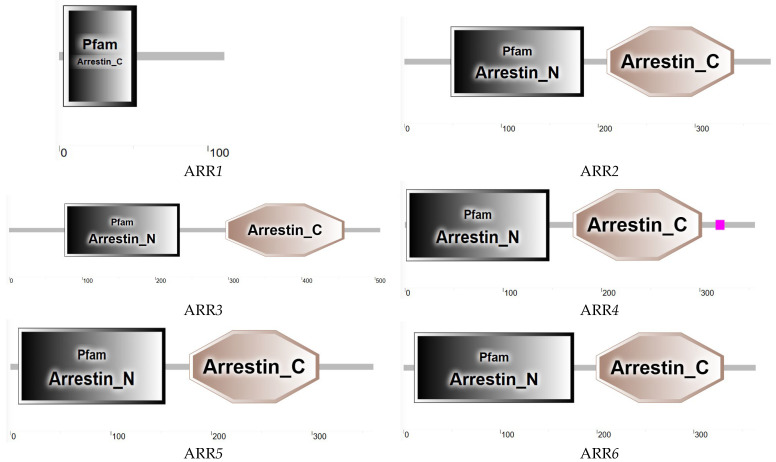
Predicted Protein Domains of ARR Gene in *T. remus*.

**Figure 8 insects-16-00393-f008:**
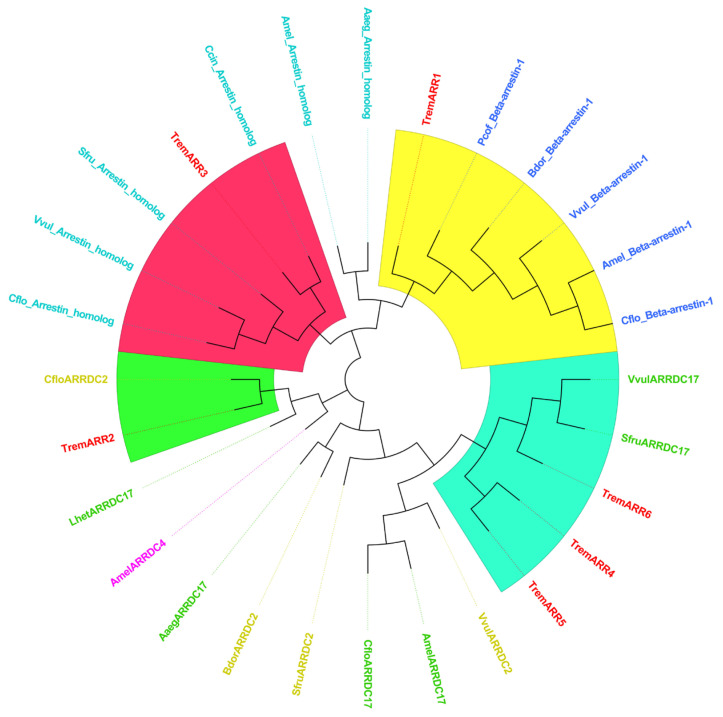
Phylogenetic Tree of ARR Gene in *T. remus*. Species include: [*Camponotus floridanus*] Beta-arrestin-1 EFN63489.1, [*Vespula vulgaris*] Beta-arrestin-1 XP_050854143.1, [*Bactrocera dorsalis*] Beta-arrestin-1 JAC46639.1, [*Phymastichus coffea*] Beta-arrestin-1 XP_058789028.1, [*Apis mellifera carnica*] ARRDC4 KAG9434969.1, [*Camponotus floridanus*] ARRDC2 XP_025267996.1, [*Vespula vulgaris*] ARRDC2 XP_050848577.1, [*Spodoptera frugiperda*] ARRDC2 XP_050561763.1, [*Bactrocera dorsalis*] ARRDC2 JAC51535.1, [*Apis mellifera*] arrestin homolog XP_016772051.1, [*Camponotus floridanus*] arrestin homolog XP_011258347.1, [*Vespula vulgaris*] arrestin homolog XP_050857736.1, [*Spodoptera frugiperda*] arrestin homolog XP_035459304.1, [*Aedes aegypti*] arrestin homolog XP_001663732.1, [*Cephus cinctus*] arrestin homolog XP_015605542.1, [*Aedes aegypti*] ARRDC17 XP_021708730.1, [*Apis mellifera*] ARRDC17 XP_001121347.2, [*Camponotus floridanus*] ARRDC17 XP_011267384.3, [*Vespula vulgaris*] ARRDC17 XP_050867923.1, [*Spodoptera frugiperda*] ARRDC17 XP_035447610.1, [*Leptopilina heterotoma*] ARRDC17 XP_043482288.1, [*Apis mellifera*] Beta-arrestin-1 XP_016769385.1.

**Figure 9 insects-16-00393-f009:**
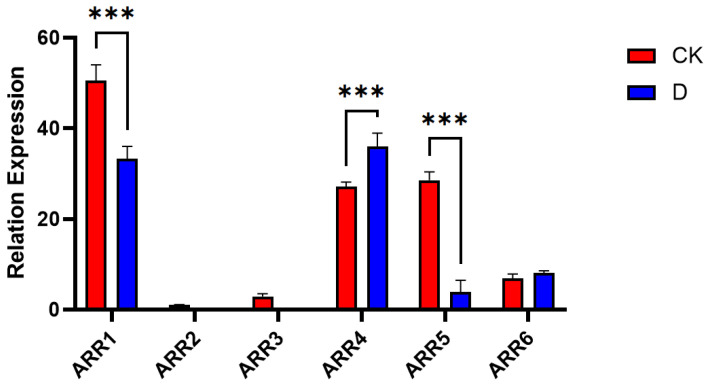
Expression Analysis of ARR Genes in Diapausing and Non-Diapausing *T. remus*. (*** represents a significant difference *p* < 0.01, “CK” refers to the normally reared *T. remus*, and “D” refers to *T. remus* that underwent diapause treatment).

**Table 1 insects-16-00393-t001:** Specific Primers for qPCR of CBP and ARR Genes in *T. remus*.

Primer	Forward Primer	Reverse Primer
RPL12	GAGGTGTGTTGGTGGAGAA	TAAGAGAGGCAGCAGAAGG
RPS2	GGGGAAACAAAATCGGTAA	CGGTGTGAGGTAGGCATAA
CBP1	ACGATTATGGAGGGTAAAGGA	ATGTCGATCAGCCTGAAATG
CBP2	GGAATGGGCTGCTTGTCTATC	CTTTCGTCGTTGGTTTCTGGT
CBP3	ATCAGTCAAGCCATCAAAGAAA	GAACGCTAGAAACTCGTCCAA
CBP4	CGCTAGTAACGAGCAGGAGA	CAGTTGTGAGAACGGACGAA
ARR1	CGCAGCGTGAGAACCTTGGC	CGTTTGAATAGCATTGGACCTTGT
ARR2	CACAAACATGACCAGGAAGC	CCAAGAAAGGTTGACGGAAG
ARR3	AACAACAGCACCAAGACCATC	TAACGAGTTTCACCCTGACATA
ARR4	ATATCCACCAGGCAACTCGG	GGCATTACCAACGCCATCAA
ARR5	AGACCCTTTAGCCAATACTCCA	TCGGCATTACCAATACCATCA
ARR6	GGCTTTAAGGAGCTGGAGAT	CTTGGTGCTTTGAGTTGTGC

**Table 2 insects-16-00393-t002:** Assembly Results of *T. remus* Transcriptome.

Length Range	Transcript	Unigene
300–500 bp	6226 (21.64%)	4074 (25.09%)
500–1000 bp	5176 (17.99%)	3876 (23.88%)
1000–2000 bp	5265 (18.30%)	3209 (19.77%)
>2000 bp	12,102 (42.07%)	5073 (31.25%)
Total Number	28,769	16,232
Total Length	72,003,323	31,186,643
N50 Length	4596	3540
Mean Length	2503	1921

**Table 3 insects-16-00393-t003:** Top 20 KEGG pathways containing the most unigenes.

KEGG Pathway	Pathway ID	Unigen Number
Ribosome	ko03010	273
Protein processing in endoplasmic reticulum	ko04141	178
Oxidative phosphorylation	ko00190	177
Endocytosis	ko04144	153
PI3K–Akt signaling pathway	ko04151	150
Lysosome	ko04142	149
Spliceosome	ko03040	144
cAMP signaling pathway	ko04024	135
MAPK signaling pathway	ko04010	120
Nucleocytoplasmic transport	ko03013	115
Ubiquitin mediated proteolysis	ko04120	115
Pancreatic secretion	ko04972	114
Retrograde endocannabinoid signaling	ko04723	113
mTOR signaling pathway	ko04150	111
AMPK signaling pathway	ko04152	111
Insulin signaling pathway	ko04910	111
Focal adhesion	ko04510	105
MAPK signaling pathway-fly	ko04013	103
Wnt signaling pathway	ko04310	101
Purine metabolism	ko00230	100

**Table 4 insects-16-00393-t004:** Candidate CBP information of *T. remus*.

Unigene Reference	ORF	Gene Name	BLAST Best Hi (Accession Number; Name; Species)	E Value	Identity	Full Length
Cluster-4246.4268	588	TremCBP1	sarcoplasmic calcium-binding protein 1 isoform X2 [Cephus cinctus]	2 × 10^−119^	81.87%	Yes
Cluster-4246.6862	1761	TremCBP2	SPARC-related modular calcium-binding protein 2 [Dufourea novaeangliae]	0	65.93%	Yes
Cluster-4246.8699	1128	TremCBP3	45 kDa calcium-binding protein [Pogonomyrmex barbatus]	3 × 10^−148^	63.11%	Yes
Cluster-4246.458	1044	TremCBP4	probable calcium-binding protein CML11 isoform X1 [Frieseomelitta varia]	3 × 10^−98^	86.29%	Yes

## Data Availability

The original contributions presented in this study are included in the article/Appendix A. Further inquiries can be directed to the corresponding author.

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
