# Peer review of "Molecular Insights into Diapause Mechanisms in Telenomus remus for Improved Biological Control"

_insects, 2025, doi:10.3390/insects16040393_

Round 1
Reviewer 1 Report
Comments and Suggestions for Authors
The manuscript by Yu et al. investigates the molecular mechanisms of diapause in Telenomus remus, an egg parasitoid used for biological control of the pest Spodoptera frugiperda. Diapause, induced by photoperiod manipulation, improves the parasitoid's survival and storage. Transcriptome analysis identified 2,642 differentially expressed genes, including those involved in calcium signaling and phototransduction. Genes such as CBP4, related to calcium signaling, and arrestin genes play key roles in diapause maintenance. The results enhance understanding of diapause regulation, offering strategies to improve the parasitoid's effectiveness as a biological control agent. This study provides novel and useful information on the diapause of the parasitoid; however, I have several major and miscellaneous suggestions for the authors' consideration to improve the manuscript.
1. The manuscript is generally not carefully prepared, containing typos and requiring rewriting. Some parts are missing (e.g., the conclusion). Additionally, I suggest the authors locate and review earlier papers on the cold storage of Telenomus spp., as well as the overwintering ecology of Telenomus spp.
2. I suggest changing the title to: "Molecular Insights into Diapause Mechanisms in Telenomus remus for Improved Biological Control."
3. Lines 23-25: Please explain the relationship of your research to sustainable agriculture. This work is highly relevant to the parasitoid’s overwintering strategy and ecology.
4. Line 27: Add the order and family of Telenomus remus and indicate that it is an important egg parasitoid.
5. Line 58: The authors should cite earlier studies on cold storage in Telenomus spp., and their claims should be based on these studies.
6. Line 63: Reference number 8 is not accessible. If it has any access option, please provide the URL or DOI of the manuscript. I suggest the authors avoid citing inaccessible studies to strengthen the scientific foundation of the study.
7. Line 68: Please indicate which Telenomus species were used in Legault et al.’s study. There are several parasitoids in the study, but reproductive diapause was detected only in one species.
8. Line 72: If T. remus overwinters as mated females, do you have any earlier studies on the seasonal ecology of T. remus?
9. Lines 76-78: Although it is stated that diapause induction is time-consuming and inefficient, such studies are crucial for elucidating overwintering success, field performance, and parasitoid ecology.
10. Line 117: Should be "diapausing females." Please rewrite as: "To terminate diapause, the diapausing females were exposed to a photoperiod of 14L:10D."
11. Line 168: Do these primers come from Telenomus remus or other organisms?
12. Lines 256-259: These statements are related to Materials and Methods.
13. Lines 265-272: This sentence is too long. Please break it into 2-3 sentences for better clarity and readability. I suggest rewriting as:
"Phylogenetic tree analysis (Figure 6) demonstrated that CBP family genes are mainly divided into four major clusters. CBP1 clusters with the sarcoplasmic calcium-binding protein (SCP) from Apis mellifera and Leptopilina heterotoma. CBP2 shows the closest relationship to the SPARC-related modular calcium-binding protein (SMOC) from Osmia lignaria. CBP3 clusters with the 45 kDa calcium-binding protein (Cab45) from Apis mellifera, and CBP4 clusters with the calmodulin (CAM) family, showing the closest relationship to Monomorium pharaonis and Apis mellifera. Previous studies have found that these four types of proteins play roles in the regulation of calcium ion-mediated light signal transduction."
14. Line 278: Please provide F and df values along with the P-value for all of your results as statistical evidence.
15. Line 334: "Drosophila" should be italicized. Please italicize all organism scientific names.
16. Lines 334-336: These sentences should be moved to the discussion section, not in the results section.
17. Figure 7: Please indicate the meaning of "" and the comparison test performed here.**
18. Line 391: Delete one of the "Diapause" repetitions.
19. Lines 397-398: Rewrite as: "Sampling of non-diapause and diapausing female adults of T. remus primarily relied on photoperiod regulation."
20. Lines 418-420: This sentence is unclear. What does "diapause treatment" refer to?
21. Lines 423-424: Rewrite as: "In diapausing individuals of Helicoverpa armigera, calcium-binding proteins were significantly downregulated."
22. Line 438: All other species names require the order and family of the organisms.
23. Line 456: The conclusion section needs to be developed further; there is currently no conclusion part.
24. Reference number 33, and all other references to Chinese journals that do not provide online access, should be replaced with references that are accessible.
Comments on the Quality of English Language
The manuscript should be edited by a native English speaker to improve readability.
Reviewer 2 Report
Comments and Suggestions for Authors
please see attachment

Round 2
Reviewer 1 Report
Comments and Suggestions for Authors
Dear Editor,
I have reviewed the revised version of the manuscript. The authors have incorporated some of my minor suggestions; however, several major comments remain unaddressed. Specifically, the following points still need attention:
- Reviewing earlier studies on the overwintering ecology of Telenomus spp. and cold storage.
- Removing inaccessible earlier studies (e.g., references 1, 2, and 3) and replacing them with papers that are accessible online in English.
In their response letter, the authors state that they added two earlier studies related to the cold storage of Telenomus spp. However, one of these references (reference number 8) is not related to Scelionids, and reference number 9 concerns the effect of host (S. litura) egg refrigeration on the parasitism of Telenomus remus. Therefore, both papers are not relevant to my comment, and the cited references are in Chinese.
I suggest that the authors use the search function of Web of Science with appropriate keywords to identify and cite relevant studies.
References:
- CHEN He-Sheng, NIU Li-Ming, FU Yue-Guan, et al. "Study on the low-temperature storage conditions of Coccophagus japonicus Compere." Journal of Environmental Entomology, 2020, 42(03): 718-724.
- LI Chuanying, CAI Yujing, SU Xiangning, et al. "Effect of Spodoptera litura eggs on parasitism in Telenomus remus under different refrigerated conditions." Guangdong Agricultural Sciences, 2024, 51(08): 1-9.
Additionally, I noticed the following issue in the manuscript:
L237: Table Error! Reference source not found...
Comments on the Quality of English LanguageI would recommend that the authors carefully review the manuscript for overall English flow and clarity. Some sections may benefit from revisions to improve readability and ensure the text is accessible to a broader audience.
Reviewer 2 Report
Comments and Suggestions for Authors
The authors applied most of the minor corrections, but avoided deeper changes. They did not change the FC, as a result of which the paper is still very general , showing the expression of many genes (up or down, Fig.3). The comparative analysis with GO database and KEGG database looks similar, where the authors decided to list the top 20 KEGG pathways. In the next stage, the authors analyze the expression of selected genes related to light signal transduction genes: calcium-binding protein (CBP) genes and arrestin (ARR) genes. In this version of the paper, they corrected the results, showing that only CBP1 and CBP2 show significant differences (line 284, Fig. 6). Unfortunately, in the discussion, the authors describe upregulated CBP4 (which according to Fig. 6 is not significant). Similarly, in the abstract (line 36-37) the role of CBP4 is described. This is incorrect and incorrectly describes the results obtained by the authors. My attention was also drawn to the fact that the new Fig 6 and Fig 9 do not show the same results as in the previous version. Therefore, the text incorrectly describes the results (line 353-357).
